# Promiscuous specialists: Host specificity patterns among generalist louse flies

**Aleksi Lehikoinen**[1�], **Pekka Pohjola**[2�], **Jari Valkama**[1�], **Marko Mutanen**[3], **Jaakko L. O. Pohjoismäki**[2]*

1 The Helsinki Lab of Ornithology, Finnish Museum of Natural History, Helsinki University, Helsinki, Finland,
2 Department of Environmental and Biological Sciences, University of Eastern Finland, Joensuu, Finland,
3 Ecology and Genetics Research Unit, University of Oulu, Oulu, Finland

☗ These authors contributed equally to this work.
* Jaakko.Pohjoismaki@uef.fi

**Data Availability Statement:** All relevant data are within the paper and its Supporting Information files. The DNA barcodes with sample details are available from the Barcode of Life database, as a public dataset of DS-FINHIPPO at dx.doi.org/10.

## Abstract

Ectoparasites such as louse flies (Diptera: Hippoboscidae) have tendency for host specialization, which is driven by adaptation to host biology as well as competition avoidance between parasites of the same host. However, some louse fly species, especially in genera attacking birds, show wide range of suitable hosts. In the presented study, we have surveyed the current status of bird specific louse flies in Finland to provide comprehensive host association data to analyse the ecological requirements of the generalist species. A thorough sampling of 9342 birds, representing 134 species, recovered 576 specimens of louse flies, belonging to six species: *Crataerina hirundinis*, *C. pallida*, *Ornithomya avicularia*, *O. chloropus*, *O. fringillina* and *Ornithophila metallica*. Despite some overlapping hosts, the three *Ornithomya* species showed a notable pattern in their host preference, which was influenced not only by the host size but also by the habitat and host breeding strategy. We also provide DNA barcodes for ten Finnish species of Hippoboscidae, which can be used as a resource for species identification as well as metabarcoding studies in the future.

## Introduction

Parasites depend on their hosts as their principal ecological niche as well as source of the essential resources [1]. Due to this intimate relationship, parasites commonly tend to specialize on the host, adapting to the host defence strategies, behaviour and ecology. Due to the selection pressures and short generation time, parasites are also prone to evolve rapidly, helping them to circumvent potential evolutionary advantages that the host has gained [2] and facilitating the specialization process. Parasitic lineages, especially endoparasites, are characterized by long branches in molecular phylogenies [3–5], for which reason they often constitute the "rogue" taxa in them. While several non-mutually exclusive explanations for this pattern have been suggested, for mitochondrial *COI* gene, this is likely at least partly explained as being an adaptation to anoxic environment [5]. Additional pressure for host specialization is driven by direct or interference competition between different parasite species occupying the same host [6]. For example, spatial segregation, which can allow the parasites to coexist on the same host can ultimately lead to intrahost speciation, as seen in *Dactylogyrus* gill parasites [7] and human lice[8].

5883/DS-FINHIPPO, including GenBank accession numbers.

**Funding:** The Finnish Barcode of Life campaign has been funded by the Kone Foundation, the Finnish Cultural Foundation, and the Academy of Finland (through FinBIF research infrastructure project). Aleksi Lehikoinen received funding from Academy of Finland (grants 323527 and 329251).

**Competing interests:** The authors have declared that no competing interests exist.

Host niches can also be partitioned temporally, as is the case with a number flea (Siphonaptera) species on small mammals, such as *Peromyscopsylla* spp. living on e.g. bank voles (*Myodes glareolus* (Schreber)) during winter months and *Ctenopththalmus* spp. during spring/summer months [9]. Parasites can also avoid competition by specializing on different aspects of the host biology. For example, the parasites can attack different developmental stages of the host, or in different biotopes or context, such as the ant decapitating scuttle flies (*Pseudacteon*, Diptera: Phoridae), where some species attack while foraging ants and some ants at the nest [10].

Louse flies (Diptera: Hippoboscidae) are obligate ectoparasites of birds and mammals, belonging to the same superfamily (Hippoboscoidea) with tsetse flies (Glossinidae). Both families are hematophagous and viviparous. As of note, bat flies (Nycteribiinae, Streblinae) have been treated as independent families, but are in fact embedded within the other Hippoboscidae taxa [11]. Adults of Hippoboscoidea species are long lived, giving birth to a full-grown or pupariated larva, which develop singly within the female's uterus, utilizing a rare mechanism known as adenotrophic viviparity [12]. Louse flies have low fecundity. Not much is known about the fecundity of bird louse flies, but a single female sheep ked (*Melophagus ovinus* (Linnaeus)) produces 12–15 [13] and tsetse flies up to eight larvae during the female's lifetime [14]. Of the 45 European species of Hippoboscidae, only 12 have been recorded in Finland [15] and of these, seven attack birds: *Crataerina hirundinis* (Linnaeus), *C. pallida* (Olivier), *Olfersia fumipennis* (Sahlberg), *Ornithomya avicularia* (Linnaeus), *O. chloropus* Bergroth, *O. fringillina* Curtis and *Ornithophila metallica* (Schiner). Three of the species are highly specialized, *C. hirundinis* on barn swallow (*Hirundo rustica* Linnaeus), *C. pallida* on common swift (*Apus apus* (Linnaeus)) and *O. fumipennis* on osprey (*Pandion haliaetus* (Linnaeus)), while the remaining four have relatively wide host range, each attacking dozens of bird species [12, 16–18]. The remaining louse flies on the Finnish check list are the bat flies *Nycteribia kolenati* Theodor & Moscona and *Penicillidia monoceros* Speiser, as well as keds *Lipoptena cervi* (Linnaeus), *Hippobosca equina* (Linnaeus) and *Melophagus ovinus* (Linnaeus). Some example specimens of Finnish louse flies species are shown in Fig 1A.

In the presented study, we sought to survey the current status of bird attacking louse flies in Finland and provide comprehensive host association data to analyse the ecological requirements of the generalist species as well as DNA barcodes for the Finnish Hippoboscidae to facilitate their identification in the future. This required the concentrated effort from 36 bird ringers, who recorded the abundance of bird specific louse flies from 9342 birds, representing 134 species. A total of 576 specimens, belonging to six species of bird flies were sampled. From these *Crataerina hirundinis* and *C. pallida*, were found only on their known specific hosts, whereas *Ornithomya avicularia*, *O. chloropus* and *O. fringillina* were found on 68 different bird species. The sixth species, *Ornithophila metallica* was represented only by one specimen. Despite some overlapping hosts, the three *Ornithomya* species showed a notable pattern in their host preference. To explain this pattern, we were interested (i) what species' traits of hosts, could explain the variation in abundance of bird flies and (ii) do species' traits of hosts differ between different generalist bird fly species. We predict that body size of host, habitat preference, migration strategy, nest location and diet could explain the variation in species and abundance of bird flies in different host species of birds. The obtained DNA barcodes work well for separating the species and can be used as a resource for species identification as well as metabarcoding studies in the future.

## Materials and methods

### Data collection and filtering

Louse fly data for the study was obtained via routine ringing of birds at different ringing stations or by local ringers, covering most of Finland (S1 Fig). The permits to catch and ring

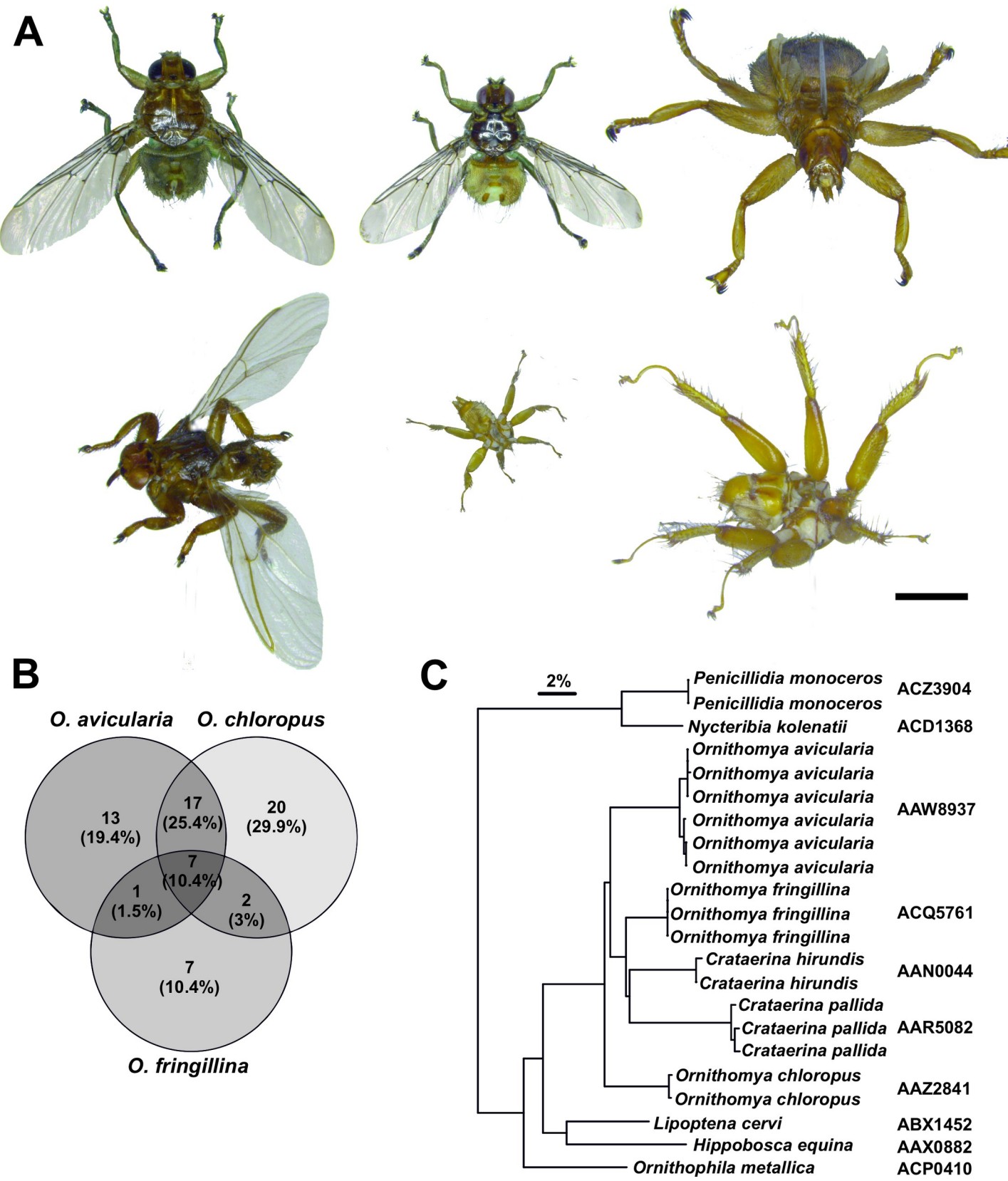

**Fig 1. Host associations and DNA barcode divergence among Finnish louse flies.** (A) Examples of different louse flies in Finnish fauna. Top row: bird louse flies *Ornithomya avicularia*, *O. chloropus* and common swift louse fly *Crataerina pallida* (note the vestigial wings). Lower row: deer ked *Lipoptena cervi*, batflies *Nycteribia kolenatii* and *Penicillidia monoceros*. All images in scale, scale bar 1 mm. (B) Host associations among the three *Ornithomya* species. Only one *Ornithophila metallica* was found in this study and *Crataerina* spp. were collected from their specific hosts, as indicated in the results. (C) Neighbor-Joining tree for the species covered in this study. Note that the tree demonstrates sequence differences between the taxa and does not represent actual phylogeny. The barcode index number (BIN) for each taxon on the right margin.

birds were issued by the Centre for Economic Development, Transport and the Environment (decision number: VARELY/3622/2017) and by the Finnish Wildlife Agency (decision number: 2018-5-600-01158-5). The data consisted two types of information: i) information if a bird has had bird fly or not and ii) information what bird fly species certain bird species had been carrying.

Voluntary bird ringers were collecting information of the bird flies when handling the bird (presence/absence). The ringers also identified the age of the bird if possible (young born during the same year or older). That data was collected during years 2008–2019, but most of the data origin from 2013 onwards when new data base system was launched allowing an easy data entry. Altogether 36 ringers participated the data collection during these years and marked information from 9342 birds (134 species 72 of which had bird flies; S1 and S2 Tables). However, only some of the ringers entered data from birds which did not show signs of louse flies and thus this data could not be used to study the prevalence among hosts. In addition, some bird ringers collected the bird flies from the birds in plastic vials with 90% ethanol for further investigation. This included altogether samples from 520 birds covering 62 bird species. We also determined the louse flies collected from injured birds treated in Korkeasaari Zoo in Helsinki. As these specimens were collected unsystematically, they were not included in the statistical analyses, but are presented in S2 Table to supplement the host records of Finnish louse fly species. All the specimen records with collection, locality and host data are uploaded to the Finnish Biodiversity Info Facility database at www.laji.fi.

We calculated prevalence of louse flies in the data of each ringer and excluded those ringers which had very high prevalence (>0.5). The aim of this filtering was to remove data from the ringers who have not actively marked zero observations, which are important for prevalence analyses. After this filtering, the data included 8352 observations (130 bird species 48 of which had had bird flies; S1 Table) collected by 13 more dedicated ringers. Each observation was classified into three different time periods: i) late spring and early summer: May and June, iii) late summer and early autumn: July-September and iii) late autumn–early spring: October-April.

## Species determination and DNA barcoding

The louse flies included in the study were determined using the relevant literature and identification keys [16, 17]. The *COI* DNA barcode region was sequenced from one to three specimens of each of the six bird louse flies collected in this study, together with other louse fly specimens, representing all but two species found from Finland (Table 1). The missing species were the sheep ked (*Melophagus ovinus* (L.)), a species that is probably close to extinction in Finland due to improved animal husbandry and veterinary practises, and the osprey specialist *Olfersia fumipennis*. DNA sequencing of the standard DNA barcode fragment of mitochondrial *COI* gene was carried out within the framework of the national campaign of Finnish Barcode of Life (https://www.finbol.org/). DNA sequencing was conducted in the Centre for Biodiversity Genomics (CGB) at the University of Guelph, Canada, following protocols outlined in deWaard et al. [19]. Briefly, DNA was isolated from the left middle leg of the specimen and the 658bp 5′region of the mitochondrial *COI* gene was amplified with LepF1 and LepR1 [20] primers. Obtained PCR product was column purified and Sanger sequenced. Details of

**Table 1. Specimens included in the DNA barcode analysis.**

| Subfamily | Species | Hosts | Country | N |
|---|---|---|---|---|
| Hippoboscinae | *Hippobosca equina* | Horse | Slovenia | 1 |
| Hippoboscinae | *Lipoptena cervi* | Moose | Finland | 1 |
| Hippoboscinae | *Crataerina hirundinis* | Barn swallow | Finland | 2 |
| Hippoboscinae | *Crataerina pallida* | Common swift | Finland | 3 |
| Hippoboscinae | *Ornithomya avicularia* | Birds | Finland | 6 |
| Hippoboscinae | *Ornithomya chloropus* | Birds | Finland | 2 |
| Hippoboscinae | *Ornithomya fringillina* | Birds | Finland | 3 |
| Hippoboscinae | *Ornithophila metallica* | Birds | Finland | 1 |
| Nycteribiinae | *Nycteribia kolenatii* | Bats | Finland | 1 |
| Nycteribiinae | *Penicillidia monoceros* | Bats | Finland | 2 |

Host preference refers to the main host taxon.

the used DNA extraction, PCR and sequencing vary due to the continuous development of the protocols of CGB, but are provided for each specimens in their sequence page and LIMS report of the Barcode of Life Data Systems (BOLD; [21]). All collection, taxonomic and sequence data as well as specimen photographs were deposited in the BOLD database and are available through the public dataset of DS-FINHIPPO at dx.doi.org/10.5883/DS-FINHIPPO, including GenBank accession numbers. Calculation of sequence divergences were conducted under Kimura 2-parameter model for nucleotide substitution and with BOLD alignment using BOLD Barcode Gap analysis tool. A Neighbor-Joining tree was built similarly under Kimura 2-parameter model in BOLD and modified with CorelDRAW 2020.

## Statistical analyses

To investigate which factors affect the abundance of bird flies in different species, we build Generalized Linear Mixed Model (GLMM) with Poisson distribution. The response variable was number of bird flies in a given bird individual. The explanatory variables were age of bird (1 = adult, 0 = unknown, -1 young), time period when the sampling was done (see the classification of the four time periods above), latitude coordinate of the record, breeding habitat class, migration strategy, nest site of the host and was the host predator or not (diet). Body size of birds was strongly correlated with the diet and was thus not included to the model. Timing and latitude were included to capture spatio-temporal variation in the abundance of flies. Age of the bird was included as e.g. body condition and timing of migration can differ between age groups [22, 23]. Breeding habitat and nest site of species was included as flies may prefer certain locations to find their hosts. The habitat classes were i) farmland, ii) forest, iii) mires and mountains, iv) scrubland and v) wetland according to Väisänen et al. [24]. The nest site classes of species were i) on land, ii) openly on trees or iii) on cavities according to Cramp et al. [25]. The reader should note that the birds were not necessarily sampled in their breeding habitats but also during the migration when the habitat type of the sampling site can differ from the breeding class. Migratory behaviour has earlier been found to affect many life-history processes of birds, such as abundance changes and moulting [23, 26]. The migration strategy classes of species were i) resident, ii) short-distance migrant (wintering mainly in Europe or Mediterranean) and iii) long-distance migrant (wintering in tropical areas) according to Saurola et al. [27] and Valkama et al. [28]. We used the diet as a variable because we expected that predator species would have higher number of flies, which may have been received from the prey species. Hawks and owls were classified as predators. Latitudes of the sampling sites were

centred before analyses. The explanatory variables did not show any clear collinearity (pearson correlation, |r |<0.32). The species was added as a random factor. Because closely related species may have similar responses due to common ancestry, we took the phylogeny of the species into account in the random structure of the model. We downloaded one phylogeny tree of the study species from www.birdtree.org [29].

The modelling was conducted using function MCMCglmm [30] in R version 3.6.0 [31] using 1,030,000 iterations, where first 30,000 were used for "burning in" and thinning interval was 1000. We used the following priors (R-structure: V = 1, nu = 0.00, G-structure: V = 1, nu = 0.02). We investigated the trace plots of the model and found the chains randomly distributed.

In the later analyses, we investigated if the species traits of the host species differ between the three main generalist bird fly species (*Ornithomya avicularia*, *Ornithomya chloropus* and *Ornithomya fringillina*). The used traits were habitat of species (see as above), migration strategy (same as above), nest site (same as above) and body mass. The habitat classes of mires and mountains (n = 3) and scrubland (n = 3) were however merged to farmland due to very small samples sizes in these groups. These three habitats formed a general open habitat type category. Each of these four variables were tested separately. The three first categorical variables were tested using chi-square (chisq.test function in R) test based on the presence or absence of the fly in a given host species in the whole data. The body mass was tested using linear regression (lm function in R), where the log-transformed mass of the host was explanatory variable and the bird fly species was explanatory variable. The data and variables used in the statistical analyses are provided in S3 and S4 Tables.

## Results

We obtained systematic data of presence/absence of louse flies on 134 bird species. A total of 576 bird fly specimens were collected by the bird ringers, representing six louse fly species (S1 and S2 Tables). *Crataerina hirundinis* (n = 2) and *C. pallida* (n = 21), were observed only from their known hosts, *Hirundo rustica* and *Apus apus*, respectively. One *Ornithophila metallica* specimen was found on spotted flycatcher (*Muscicapa striata* (Pallas)) captured for ringing in Siikajoki, June 4, 2011. This is the second record for the species from Finland. The remaining three generalist species, *Ornithomya avicularia* (n = 105), *O. chloropus* (n = 339) and *O. fringillina* (n = 108), showed considerable variation in their host preference, totalling 67 different bird species, when the host records from Korkeasaari zoo bird shelter are taken into account (S1 and S2 Tables, Fig 1B).

The abundance these generalist louse flies (from 0 to 5) was explained by habitat of the species, predatory class, time period (Table 2) and latitude. Species breeding in mires and mountains had significantly fewer bird flies than species breeding in farmlands, and there was also similar tendency in birds breeding in wetlands. Predators had significantly higher number of bird flies than non-predatory species. Bird flies were more abundant in July-September period compared to May-June period, whereas abundances were smaller during October-April (Table 2). Abundances of flies also increased slightly with increasing latitude (Table 2).

Host species of *O. fringillina* (mean 14 g) had clearly smaller body size than hosts of *O. avicularia* (mean 311 g; $t$ = -4.00, $p < 0.001$), but interestingly hosts of *O. chloropus* (mean 235 g) did not differ from *O. avicularia* ($t$ = -0.90, $p$ = 0.368), although the latter has been generally associated with larger hosts. The breeding habitats of hosts also differed significantly between louse fly species ($\chi^2$ = 10.99, $df$ = 4, $p$ = 0.027; Table 3). *Ornithomya fringillina* avoided hosts that were breeding in open habitat types, but were preferring hosts breeding in forest habitats, whereas opposite was the case in *O. chloropus*. All three bird fly species tend to avoid hosts

**Table 2. Parameter estimates and P-values of the model explaining abundances of bird flies in different bird species.**

| Variable | Posterior estimate (min, max) | *p*-value |
|---|---|---|
| **(Intercept)** | **-4.57 (-6.74, -2.55)** | **<0.001** |
| Age (adults compared to young) | -0.08 (-0.22, 0.05) | 0.244 |
| Habitat, forest | -0.22 (-1.12, 0.58) | 0.558 |
| **Habitat, mires and mountains** | **-2.68 (-4.18, -0.96)** | **<0.001** |
| Habitat, scrubland | -0.42 (-1.95, 0.99) | 0.550 |
| Habitat, wetland | -0.96 (-2.07, 0.34) | 0.098 |
| **Predator (compared to non-predator)** | **2.26 (0.12, 4.28)** | **0.042** |
| Migration, resident | 0.23 (-0.80, 1.23) | 0.660 |
| Migration, short-distance migrant | -0.04 (-0.82, 0.86) | 0.942 |
| Nest site, land | -0.01 (-0.99, 0.97) | 0.958 |
| Nest site, openly on trees | 0.30 (-0.76, 1.43) | 0.560 |
| **Time, Jul-Sep** | **0.89 (0.58, 1.26)** | **<0.001** |
| **Time, Oct-Apr** | **-1.64 (-2.32, -1.01)** | **<0.001** |
| **Latitude** | **0.11 (0.04, 0.17)** | **<0.001** |

Age refers to the host age. Habitat classes were compared to hosts breeding farmlands. Migration strategy was compared to long-distance migratory hosts. Nest sites were compared to hosts breeding cavities. Time period was compared to situation in May-June. Latitude was centred decimal coordinate of the data collection site. Significant (*p*<0.05) variables are bolded.

breeding in wetland habitats. There was also a tendency that nest site of birds would explain host species selection of different bird fly species ($\chi^2 = 8.38$, *df* = 4, *p* = 0.079; Table 4). *Ornithomya avicularia* tend to have more often hosts breeding openly on trees and avoidance for species breeding on the ground, whereas opposite was the case in *O. chloropus*. *O. fringillina* showed weak preference towards host species breeding in cavities and avoidance towards species breeding openly on trees. The migratory behaviour of hosts did not differ between bird fly species ($\chi^2 = 5.29$, *df* = 4, *p* = 0.259; Table 5).

Sequencing of DNA barcode fragment of COI gene indicated all included ten louse fly species having a highly distinct DNA barcode (Fig 1C). The single specimen of the sheep ked (*Melophagus ovinus*) analyzed by us failed to yield any sequence data, but public BOLD records of it indicate it also having a distinct barcode as well. Therefore, of Finnish louse flies, only rarely encountered *Olfersia fumipennis* fully lacks the barcode information in the BOLD reference library. The mean of minimum genetic divergence between the species was 8.34% and at minimum, the two species differed from each other by 6.24% (*Ornithomya hirundinis* vs. *O. fringillina*). While intraspecific variability could not be assessed for four species as represented by singletons only, it never exceeded 1%. Overall, this result suggest a wide barcode gap to exist between the Finnish louse flies. All species also were assigned to their own BINs (Barcode Index Number) as well.

**Table 3. Observed and expected (in brackets) number of host species breeding in forest, open habitats and wetlands in three bird fly species.**

| Species | Forest | Open | Wetland |
|---|---|---|---|
| *O. avicularia* | 22 (21.9) | 7 (8.4) | 6 (4.7) |
| *O. chloropus* | 23 (28.1) | 16 (10.8) | 6 (6.0) |
| *O. fringillina* | 15 (10.0) | 0 (3.8) | 1 (2.1) |

**Table 4. Observed and expected (in brackets) number of host species breeding in cavities, on ground and openly on trees in three bird fly species.**

| Species | Cavity | Ground | Trees |
|---|---|---|---|
| *O. avicularia* | 7 (8.0) | 12 (15.3) | 16 (11.7) |
| *O. chloropus* | 8 (10.3) | 24 (19.7) | 13 (15.0) |
| *O. fringillina* | 7 (3.7) | 6 (7.0) | 3 (5.3) |

## Discussion

Host-parasite coevolution pushes parasites to specialize by adapting them to the host defence mechanisms and ecological niche [2]. As an additional factor, competition between parasites of the same host can further drive niche specialization within and between hosts [1, 6]. Louse flies are obligate ectoparasites, many of which show considerable specialization to single or few hosts. In general, wingless or short-winged (stenopterous) species of louse flies are highly specialized, including the swift and swallow parasites of the genus *Crataerina*. In contrast, the species of *Ornithomya* have fully developed wings and many of the known species have relatively broad host range. Compared to the more specialized winged louse flies, such as *Lipoptena*, the *Ornithomya* species are also active fliers, which could be and adaptation to short lived or otherwise risky host niche. Ability to change host individual combined with flexibility with the host species is likely to be a part of risk avoidance strategy. Unlike with most other parasitic insects, such as fleas, whose larvae occupy completely different niche as detritus-feeders [9], the survival of the female louse fly and its offspring is coupled to the extreme. As the female louse fly nurtures only one larva at the time, the number of produced offspring increases with the longevity of the female and is unparallel to most insects, where the adult stage is ephemeral compared to the larval stage, and number of the offspring as well as their mortality is large.

The purpose of our survey of bird parasitic louse flies was twofold. The first was to provide a systematic overview of the current status of the fauna, including the monitoring of potential range expansion of species under the current climate change. In comparison, in central Europe alone there are twice as many species of bird infesting louse flies than have been recorded from Finland [15, 32]. We were able to sample all bird louse fly species previously known from Finland, except for the osprey specialist *Olfersia fumipennis*. Disappointingly, the only louse fly collected from an osprey was *O. avicularia* (S2 Table). The last record of *O. fumipennis* from Finland is in fact the type specimens from 1884, which would qualify it as regionally extinct. However, because ospreys are not uncommon in Finland, *O. fumipennis* might be possible to rediscover by more systematic search. As no conclusions about the species current status can be drawn, *O. fumipennis* is listed as DD in the latest Finnish Red List [33].

Interestingly, also no new species to Finland were recovered among the sampled 576 louse fly specimens. For example, we checked carefully all *Ornithomya* specimens collected from barn swallows as *Ornithomya biloba* Dufour, a barn swallow specialist, is present in neighbouring Sweden [16], but these all turned out to be the common *O. avicularia* or *O. chloropus* (S2 Table). Similarly, migratory birds frequently transport louse fly species with widespread

**Table 5. Observed and expected (in brackets) number of host species based on migratory strategy in three bird fly species.**

| Species | Long | Short | Resident |
|---|---|---|---|
| *O. avicularia* | 13 (13.1) | 16 (13.9) | 6 (8.0) |
| *O. chloropus* | 19 (16.9) | 17 (17.8) | 9 (10.3) |
| *O. fringillina* | 4 (6.0) | 5 (6.3) | 7 (3.7) |

southern or cosmopolitan distribution, such as *Pseudolynchia canariensis* (Macquart) or *Ornithoica turdi* (Latreille). The only such example was a single *Ornithophila metallica* specimen was found on spotted flycatcher, representing the second record for this Ethiopian-Oriental species from Finland. Some louse fly species would require targeted effort to discover. For example, the grey heron (*Ardea cinerea* Linnaeus) has become relatively common in southern Finland during the past two decades and is a host for *Icosta ardeae* (Macquart).

The second goal of the survey was to obtain comprehensive host data for the common generalist *Ornithomya* species and use it to dissect the ecological requirements of the different species. Despite the wide and overlapping host ranges among *Ornithomya*, a general pattern of host preference has been known to exist between the different species [16, 32]. For example, *O. fringillina* is almost unexceptionally found only on small host birds. The question of host preference is naturally complicated by the fact that association of a mobile louse fly species on a bird species does not indicate a true host-parasite relationship. Predatory birds are likely to obtain parasites from their prey and the flies might probe several false candidates in search for their specific host. In fact, this was the case in our study as well, where the predatory birds had significantly larger numbers of louse flies (Table 2). However, our analysis reveals some general patterns of host bird association among the Finnish *Ornithomya* species (Tables 2–4, Fig 1B) Notably, *O. avicularia* prefers largest, tree breeding host bird species, whereas *O. chloropus* attacks similar sized ground breeding hosts in open habitats. In contrast, *O. fringillina* tend to prefer small, cavity breeding forest birds. Overall species breeding in northern open habitats had least number of flies, although the louse fly prevalence in generally increased towards north with the peak time for the flies being late summer (Table 2). Apart for the *Ornithophila metallica*, all observed species can be considered residential in Finland, overwintering as puparia and attacking the birds during the summer season, regardless of their migratory status (Table 5).

DNA barcodes work well for the louse flies and the sequence differences between the taxa are markedly big (Fig 1C). No cases of barcode sharing between species were detected, and despite rather scarce genetic sampling, it appears very unlikely given the wide gap between intra- and interspecific variation. This observation suggests that DNA barcoding provides as an accurate tool to identify species of louse flies. As of note, Hippoboscidae remain scarcely sampled in the BOLD (https://www.boldsystems.org/), probably because they are highly specialized and usually only found if specifically searched from their hosts. For example, at the writing of this manuscript there is only one *Ornithophila metallica* sample in the database from South Africa, which matches the Finnish specimen 98.9%. Reference DNA barcodes not only provide a determination tool for non-specialist, but also facilitate modern biodiversity surveys, such as metabarcoding studies. As an example, it was possible to detect rarely observed bat louse fly *Nycteribia kolenatii* among multiple prey species of Daubenton's bat (*Myotis daubentonii* (Kuhl)) in a study analyzing the diet of the bats from fecal DNA [34]. One aspect that we could not reliably assess is whether the generalist species of *Ornithomya* could be with cryptic specialists 'hiding' among them. Our sampling does not suggest this being the case, but we included only 3–6 specimens of the generalist *Ornithomya*, which is too little to assess this possibility confidently. Other studies have demonstrated putative generalist parasitic insects actually comprising many morphologically highly similar species of generalists [35–39]. Further studies are likely to reveal a plenty of cases of cryptic diversity among seemingly generalist species.

We conclude that although some species can be targetedly searched, considerable effort is needed to survey louse fly fauna and most new species are found by accident. Despite their wide host ranges, the different *Ornithomya* species show clear pattern of specialization to host biology and biotope, which is likely to result from competition avoidance. DNA barcodes

work well for Hippoboscidae and there are considerable distances between taxa, as is typical for parasites.

## Supporting information

**S1 Fig. Map of the bird ringing sites in Finland, which contributed to the data collection.**
(DOCX)

**S1 Table. List of 134 bird species which had been investigated by the bird ringers.** The number investigated individuals per species (N) and number of parasitized hosts per species (Flies) are reported from all investigated birds (all) and only those birds which had been systematically sampled by ringers (syst). The prevalence of host species has been calculated based on systematic surveys (- = is no data) and values where less than 10 birds have been sampled are shown in brackets due to small sample sizes.
(DOCX)

**S2 Table. Occurrences of six bird fly species (C. hir = *Crataerina hirundinis*, C. pal = *Crataerina pallida*, O. avi = *Ornithomya avicularia*, O. chl = *Ornithomya chloropus*, O. fri = *Ornithomya fringillina*, O. met = *Ornithophila metallica*) according to their observed host bird species.**
(DOCX)

**S3 Table. Bird louse fly counts vs. host species, geographical location and different ecological variables of the host.**
(CSV)

**S4 Table. Bird louse fly species vs. host species, size and ecology.**
(CSV)

## Acknowledgments

We are indebted to the numerous voluntary bird ringers, whose help and commitment made this study possible. We would like to than Eero Vesterinen, Thomas Lilley and Dr. Tomi Trilar for providing the Nycteribiinae and *Hippobosca equina* specimens for the DNA barcoding. We are also grateful to the staff of Centre for Biodiversity Genomics (University of Guelph, Canada) for continuous support with barcoding samples of the FinBOL initiative.

## Author Contributions

**Conceptualization:** Aleksi Lehikoinen, Jaakko L. O. Pohjoismäki.

**Data curation:** Pekka Pohjola, Jaakko L. O. Pohjoismäki.

**Formal analysis:** Aleksi Lehikoinen, Pekka Pohjola, Marko Mutanen, Jaakko L. O. Pohjoismäki.

**Funding acquisition:** Marko Mutanen.

**Investigation:** Aleksi Lehikoinen, Pekka Pohjola, Jari Valkama, Jaakko L. O. Pohjoismäki.

**Methodology:** Aleksi Lehikoinen.

**Project administration:** Jaakko L. O. Pohjoismäki.

**Resources:** Jaakko L. O. Pohjoismäki.

**Supervision:** Marko Mutanen, Jaakko L. O. Pohjoismäki.

**Validation:** Jaakko L. O. Pohjoismäki.

**Visualization:** Jaakko L. O. Pohjoismäki.

**Writing – original draft:** Aleksi Lehikoinen, Marko Mutanen, Jaakko L. O. Pohjoismäki.

**Writing – review & editing:** Aleksi Lehikoinen, Marko Mutanen, Jaakko L. O. Pohjoismäki.

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
