## [Decision Letter · Decision Letter 0]

15 Apr 2021

PONE-D-21-04570

Promiscuous specialists: Host specificity patterns among generalist louse flies

PLOS ONE

Dear Dr. Pohjoismäki,

Thank you for submitting your manuscript to PLOS ONE. After careful consideration, we feel that it has merit but does not fully meet PLOS ONE’s publication criteria as it currently stands. Therefore, we invite you to submit a revised version of the manuscript that addresses the points raised during the review process.

Two reviewers have provided comments on your manuscript. They are both positive but asked minor revisions.

Please read carefully the PLOS Data policy before submitting a new version.

We look forward to receiving your revised manuscript.

Kind regards,

Pierfilippo Cerretti, Ph.D.

Academic Editor

PLOS ONE

Journal Requirements:

In your Methods section, please provide additional location information of the ringing sites, including geographic coordinates for the data set if available.

In your Methods section, please provide additional information regarding the permits you obtained for the work. Please ensure you have included the full name of the authority that approved the ringing activities and ringing sites access, if no permits were required, a brief statement explaining why.

We note that you have stated that you will provide repository information for your data at acceptance. Should your manuscript be accepted for publication, we will hold it until you provide the relevant accession numbers or DOIs necessary to access your data. If you wish to make changes to your Data Availability statement, please describe these changes in your cover letter and we will update your Data Availability statement to reflect the information you provide.

Thank you for stating the following in the Acknowledgments Section of your manuscript:

Eero Vesterinen, Thomas Lilley for Nycteriibae. Ringers. Dr. Tomi Trilar for Hippobosca equina specimens. We are grateful to the Finnish Barcode of Life campaign and its funders since 2011, namely the Kone Foundation, the Finnish Cultural Foundation, and the Academy of Finland (through FinBIF research infrastructure project), for making sequencing analyses possible. We are also grateful to the staff of Centre for Biodiversity Genomics (University of Guelph, Canada) for continuous support with barcoding samples of the FinBOL 348 initiative. AL received funding from Academy of Finland (grants 323527 and 329251).

The authors received no specific funding for this work.

Please include captions for your Supporting Information files at the end of your manuscript, and update any in-text citations to match accordingly. Please see our Supporting Information guidelines for more information: http://journals.plos.org/plosone/s/supporting-information.

Reviewers' comments:

Reviewer's Responses to Questions

**Comments to the Author**

1. Is the manuscript technically sound, and do the data support the conclusions?

Reviewer #1: Yes

Reviewer #2: Yes

2. Has the statistical analysis been performed appropriately and rigorously? 

Reviewer #1: Yes

Reviewer #2: Yes

3. Have the authors made all data underlying the findings in their manuscript fully available?

Reviewer #1: Yes

Reviewer #2: No

4. Is the manuscript presented in an intelligible fashion and written in standard English?

Reviewer #1: Yes

Reviewer #2: Yes

5. Review Comments to the Author

Reviewer #1: This manuscript reports a 576 specimens of louse flies, belonging to six species: Crataerina hirundinis, C. pallida, Ornithomya avicularia, O. chloropus, O. fringillina and Ornithophila metallica, found on 9342 birds belonging to 134 species. Authors also provide DNA barcodes and louse flies host preference patterns. Generaly complexity of generalists-specialists is a frequently discussed topic in Hippboscids and it is not easy to answer. Although the article does not respond 100 percent to one of these answers, it nevertheless provides many valuable clues. As a result, it is a very valuable source of information that has been very well analyzed by the authors.

Given this report and the literature, I believe this to be an interesting manuscript and important to the readers in the field of parasitology, ecology, ornithology, wildlife and entomology. Also, the article may have an important impact and may draw attention to the problem of hippoboscids in migratory birds. English language is proper for such studies. Based on my experience statistics and molecular studies, are performed properly, and overall methodology is valid. The results are presented correctly. In particular, the discussion has been written extremely exhaustively and shows very well the important role of this research. In several points, the introduction and methodology requires reflection

Overall the article is very well written, but there are some minor errors that need to be resolved.

The most important disadvantages of the article are listed below:

Major flaws:

- The end of the introduction contains a lot of information that, in my opinion, should be found in other sections of the article (e.g. results or materials and methods). Some of the methodology statements should be included in the discussion. I would very much like the authors to look at the text again and to distribute it correctly. I have marked the most important ones in the attached file.

- All statistical abbreviations should be written in italics.

Minor flaws:

-In the attachment I am sending an article with suggestions that require reflection.

With revision addressing these concerns, I would recommend this manuscript for publication.

Reviewer #2: The authors of the manuscript “Promiscuous specialists: Host specificity patterns among generalist louse flies” have accumulated comprehensive host association dataset of bird attacking louse flies in Finland to understand the ecological requirements of generalist species of the group. The sample sizes are very impressive and the study has adequately addressed patterns that could explain the species variation and abundances of hippoboscid flies in different host species. The authors have also provided DNA barcodes of a majority of the Finnish species of louse flies which is an important resource for future quick and precise identifications of louse flies. The study is however limited to a single country but does provide the baseline for understanding host specific patterns in an enigmatic fly group.

Here are some general comments on the manuscript are below. I’ve also included some comments/edits on the pdf.

-The “Species determination and DNA barcoding” method section needs more details. Specifics on the primers, length of barcode used and sequencing platform used to generate the barcode sequence. Was there any post processing of raw data before the calculation of sequence divergences and NJ tree building?

-Statistical analyses: What is the motivation or criteria used when deciding the parameters here (explaining abundances of bird flies in different bird species). Is this a general suite generally used for such studies? Any other factors that could not be considered due to lack of data? Also I could not find the raw data for this. This data needs to be shared as part of the journal requirements.

-It would be nice to see some images of representative flies in the manuscript especially with their interesting morphology.

-It was not clear to me why promiscuous is used in the title.

6. PLOS authors have the option to publish the peer review history of their article (what does this mean?). If published, this will include your full peer review and any attached files.

Reviewer #1: No

Reviewer #2: No

---

## [Author Response · Author response to Decision Letter 0]

27 Apr 2021

Response to the reviewers

Reviewer #1:

R1: Given this report and the literature, I believe this to be an interesting manuscript and important to the readers in the field of parasitology, ecology, ornithology, wildlife and entomology. Also, the article may have an important impact and may draw attention to the problem of hippoboscids in migratory birds. English language is proper for such studies. Based on my experience statistics and molecular studies, are performed properly, and overall methodology is valid. The results are presented correctly. In particular, the discussion has been written extremely exhaustively and shows very well the important role of this research. In several points, the introduction and methodology requires reflection

A: First of all - thank you for your kind words. We are pleased to hear that that the reviewer finds louse fly ecology, a rather specialized aspect in biodiversity research, interesting.

R1:The end of the introduction contains a lot of information that, in my opinion, should be found in other sections of the article (e.g. results or materials and methods). Some of the methodology statements should be included in the discussion. I would very much like the authors to look at the text again and to distribute it correctly. I have marked the most important ones in the attached file.

A: We feel that the introduction needs a short explanation of the work. We hope that the reviewer is satisfied with the compromise we have tried to now make here. Similarly, L156-9: We feel that this is important information for the data included in the GLMM and therefore should be mentioned here and not in the discussion.

R1:All statistical abbreviations should be written in italics.

A: Corrected, thank you.

R1:In the attachment I am sending an article with suggestions that require reflection.

A: We have now made the suggested changes, apart for the suggestions for:

L67: Parentheses around the author names are used only when the genus name has been changed since the original description of the species. For example, Olfersia fumipennis was originally described as Lynchia fumipennis, Sahlberg but W. Dale transferred the species to the genus Olfersia in 1969 and therefore the author is written in parenthesis.

L134: We understand the confusion and now specify that Melophagus ovinus is almost extinct from Finland, but not globally.

L156-9: We feel that this is important information for the data included in the GLMM and therefore should be mentioned here and not in the discussion.

Reviewer #2: 

R2:The authors of the manuscript “Promiscuous specialists: Host specificity patterns among generalist louse flies” have accumulated comprehensive host association dataset of bird attacking louse flies in Finland to understand the ecological requirements of generalist species of the group. The sample sizes are very impressive and the study has adequately addressed patterns that could explain the species variation and abundances of hippoboscid flies in different host species. The authors have also provided DNA barcodes of a majority of the Finnish species of louse flies which is an important resource for future quick and precise identifications of louse flies. The study is however limited to a single country but does provide the baseline for understanding host specific patterns in an enigmatic fly group.

R2:Here are some general comments on the manuscript are below. I’ve also included some comments/edits on the pdf.

A: Noted, thank you. We have made the suggested edits. Referring to table S3 was an accidental leftover from a previous manuscript version. Note that we have now added different tables S3 and S4, which contain the datasets used in the statistical analysis.

R2:The “Species determination and DNA barcoding” method section needs more details. Specifics on the primers, length of barcode used and sequencing platform used to generate the barcode sequence. Was there any post processing of raw data before the calculation of sequence divergences and NJ tree building?

A: Details of the COI sequencing added. DNA barcoding is often done in a connection of national and international barcoding initiatives, meaning that a variety of protocols have been used to generate sequences. This is true also in our case, but for those generated in the Canadian Centre for DNA Barcoding, each record’s sequence page includes a LIMS report that provides details of the protocols used. A mention of this is now added.

R2:Statistical analyses: What is the motivation or criteria used when deciding the parameters here (explaining abundances of bird flies in different bird species). Is this a general suite generally used for such studies? Any other factors that could not be considered due to lack of data? Also I could not find the raw data for this. This data needs to be shared as part of the journal requirements.

A: We thank the referee for this comment. We have now clarified the motivation of the used parameters in the analyses. There are in general very common species traits in avian bird analyses, which we also predicted that could have also ecological importance here. 

R2:It would be nice to see some images of representative flies in the manuscript especially with their interesting morphology.

A: A nice idea. We have now provided some species photos in the figure 1.

R2:It was not clear to me why promiscuous is used in the title.

A: The title is meant to communicate a paradox: in our study we have three louse fly species with wide host range (i.e. they are promiscuous in their host use). However, when looked in detail, the species are specialized to their corresponding host niches.

---

## [Editor Report · Decision Letter 1]

18 May 2021

Promiscuous specialists: Host specificity patterns among generalist louse flies

PONE-D-21-04570R1

Dear Dr. Pohjoismäki,

We’re pleased to inform you that your manuscript has been judged scientifically suitable for publication and will be formally accepted for publication once it meets all outstanding technical requirements.

Kind regards,

Pierfilippo Cerretti, Ph.D.

Academic Editor

PLOS ONE
---

## [Editor Report · Acceptance letter]

20 May 2021

PONE-D-21-04570R1 

Promiscuous specialists: Host specificity patterns among generalist louse flies 

Dear Dr. Pohjoismäki:

I'm pleased to inform you that your manuscript has been deemed suitable for publication in PLOS ONE. Congratulations! Your manuscript is now with our production department. 

Kind regards, 

on behalf of

Dr. Pierfilippo Cerretti 

Academic Editor

PLOS ONE